# Revisiting the Common Practice of Sellars and Tegart's Hyperbolic Sine Constitutive Model

**Soheil Solhjoo**

ENTEG, Faculty of Science and Engineering, University of Groningen, Nijenborgh 4, 9747 AG Groningen, The Netherlands; soheil@solhjoo.com

**Abstract:** The Sellars and Tegart's hyperbolic sine constitutive model is widely practiced in describing stress–strain curves of metals in hot deformation processes. The acceptance of this phenomenological model is owed to its versatility (working for a wide range of stress values) and simplicity (being only a function of strain, strain rate, and temperature). The common practices of this model are revisited in this work, with a few suggestions to improve its results. Moreover, it is discussed that, with the progress of data-driven models, the main reason for using the Sellars and Tegart's model should be to identify reliable activation energies, and not the stress–strain curves. Furthermore, a piece of code (*Hot Deformation Fitting Tool*) has been created to automate the analysis of stress–strain curves with various models.

**Keywords:** hot deformation; constitutive modeling; phenomenological models; stress–strain curve; activation energy

## 1. Introduction

The attempts on modeling stress–strain curves at high temperatures can be divided into three categories: physical, phenomenological, and data-driven models; see, e.g., [1] for a short review on this topic. Data-driven models, such as artificial neural networks (ANNs), are powerful tools to build digital twins of components with complex nonlinear behaviors [2], although they do not provide physical insights into the modeled phenomena. Building accurate ANNs, however, has some difficulties: there is neither a general structure ensuring the accuracy of the trained network nor a standard procedure to design a reliable network's architecture; research on overcoming such issues is on going [3–6]. (The current work is not mainly to discuss ANNs; yet, some limited results are presented in Appendix B).

Although the inability of data-driven models to provide physical insights may be neglected in industrial applications, it is their main drawback for investigating deformation's physical mechanisms. For this purpose, physical and phenomenological models are better choices. These models are based on certain, and probably limiting, assumptions, such as simplified models for describing dislocation density and mobility (e.g., Mecking-Kocks [7,8], Estrin-Mecking [9], Zerilli-Armstrong [10], Svyetlichnyy [11,12], Voyiadjis-Almasri [13]), (non-)linear dependence of the work hardening rate on stress [14,15] and strain [1,16], or smooth transition in the stress by the domination of dynamic recrystallization [17–19]. Moreover, some of these models can be used as analytical tools to identify the critical strain for the onset of dynamic recrystallization [16,20–23]; interested readers are referred to [24] for a detailed review. Some of these physical and phenomenological models are self-consistent, meaning that all of their parameters can be identified directly from the experimental data, e.g., Johnson–Cook [25], Khan–Huang [26], and Sellars–Tegart [27]; however, most of these models are functions of certain characteristic data, e.g., initial stress, peak/steady-state stress and their corresponding strain, and deformation's activation energy, requiring separate self-consistent models to provide such data, among which the Sellars–Tegart's hyperbolic sine (ST) model [27] is the most practiced one for this purpose.

The current work discusses the deficiency of the ST model and its common practices. The ST model and its common calibration method are shortly described. Moreover, the model is calibrated for an aluminum-based nanocomposite. Then, upon discussing the results, a few suggestions are given for overcoming the considered assumptions. The aim is not simply to increase the accuracy of the model, because the data-driven models would easily surpass that. Considering that the only physically interpretable parameter of the ST model is the deformation's activation energy, this paper tries to make the model a relevant analytical tool for extracting reliable and physically meaningful activation energies.

## 2. A Brief Overview of the ST Model

As is described by Jonas et al. [28], after careful examinations of available hot-working data, Sellars and Tegart found that the steady-state data can be divided into two categories to be modeled separately:

$$Z = A' \sigma^{n'} \qquad \text{for low stresses and} \qquad (1a)$$

$$Z = A'' \exp(\beta\sigma) \qquad \text{for high stresses} \qquad (1b)$$

where $A'$, $A''$, $n'$ and $\beta$ are constants, $\sigma$ is stress and $Z = \dot{\varepsilon}\exp(Q/RT)$ is the Zener–Hollomon parameter [29], with $\dot{\varepsilon}$, $Q$, $R$ and $T$ being strain rate, activation energy, ideal gas constant, and temperature, respectively. Comparing the similarities between the steady conditions for both creep and hot working, Sellars and Tegart proposed to use the Garofalo's creep equation [30] as a more general relation between $Z$ and $\sigma$ covering a wide range of stresses [27]:

$$Z = A(\sinh(\alpha\sigma))^n \qquad (2)$$

where $A$, $\alpha$, and $n$ are constants. One can fit the model to the experimental data using optimization techniques [31,32]; however, the most common method of calibrating the model is the conventional 6-step procedure, summarized in Table 1. It is discussed that $\alpha$ can be obtained only as an approximate value [33]; therefore, to increase the accuracy of the model, its value should be varied within a certain range, and the value resulting in the best fit should be selected, e.g., see [34,35]. The common method, however, is to calculate it as $\alpha = \beta/n'$ (see Appendix A.1). Once $\alpha$ is in hand, the other constants of Equation (2) can simply be determined, although this cannot be performed by applying both temperature and strain rate simultaneously [33]. Moreover, to make the approximated stress as a function of strain, the constants of the model are considered functions of strain as well and mostly in the form of polynomials.

**Table 1.** The conventional 6-step procedure for estimating the parameters of Equation (2) at any given strain.

| No. | Parameter | Description | X | Y |
|-----|-----------|-------------|---|---|
| 1 | $n'$ | slope [a] | $\ln(\sigma)$ | $\ln(\dot{\varepsilon})$ |
| 2 | $\beta$ | slope | $\sigma$ | $\ln(\dot{\varepsilon})$ |
| 3 | $\alpha$ | $= \beta/n'$ | — | — |
| 4 | $n$ | slope | $\ln(\sinh(\alpha\sigma))$ | $\ln(\dot{\varepsilon})$ |
| 5 | $Q$ | slope | $1/RT$ | $n\ln(\sinh(\alpha\sigma))$ |
| 6 | $A$ [b] | intercept [c] | $1/RT$ | $n\ln(\sinh(\alpha\sigma))$ |

[a] In calculation of slopes ($\Delta Y/\Delta X$), it is common to assume a linear relation between X and Y over all data points. [b] Commonly, coefficient $A$ is expressed as $\ln(A)$. [c] $\ln(A)$ can be expressed in terms of intercept ($y_0$) and strain rate as $\ln(A) = y_0 - \ln(\dot{\varepsilon})$.

Some research discussed that the model may fail due to the high nonlinearity of the hot deformation behavior and the high dependency of the model's parameters on strain [36]; however, the majority of the investigations stand in favor of the model. Due to the versatility of this model, it has become a common practice within the metal forming

community to use it as a tool for analyzing and modeling the stress–strain curves obtained from hot deformation processes.

## 3. Determining the Constants of the Model

In this section, the 6-step procedure for finding the constants of the ST model (Table 1) are taken to describe the stress–strain curves of an aluminum-based nanocomposite, in which its data are collected from the literature [37]. Moreover, a piece of code, called *Hot Deformation Fitting Tool* (HDFT), is developed and written in MATLAB 9 exclusively for performing the calculations of this study; the code is published under the terms of the GNU GPLv3 license and is available at https://github.com/soheilsolhjoo/HDFT (accessed on 11 November 2021) [38].

The findings of the fitting procedure are reported in the rest of this section. The determined constants are averaged and described by a polynomial of degree 9 as functions of strain $P^9(\varepsilon)$, with $P^h(x) = \sum_{i=0}^{i=h} p_i x^i$, where $p_i$ are constants. In all of those reported figures that show an average data set, the data points and the fitted polynomial are depicted with black circles and a black line, respectively. Moreover, a strain of 0.05 is arbitrarily selected to demonstrate those plots that refer to a certain value of strain.

### 3.1. Finding $n'$

Figure 1a shows the plots used in obtaining $n'$. Because $n'$ is defined to be temperature-independent, an average of the slopes of all worked temperatures is assumed to be the value of $n'$ at any given strain. Figure 1b shows that $n'$ depends on both strain and temperature; however, it is conventionally described as $n' = f(\varepsilon)$ using its temperature-averaged values, regardless of the dependence of $n'$ on temperature.

Despite the above-mentioned common practice, let us assume a single value of $n' \approx 7.5$ for the current study, averaged over all investigated temperatures and strains. In here, the strain data points are equispaced, and to reach an acceptable approximation, a fine discretization of $\Delta\varepsilon = 0.005$ is used.

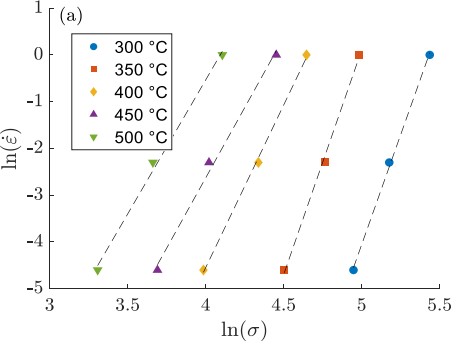 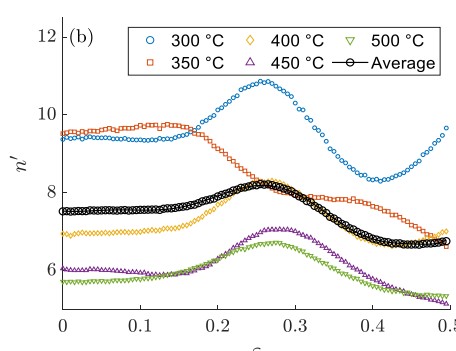

**Figure 1.** (**a**) The linear plots for determining $n'$, and (**b**) the values of $n'$ and their temperature-averaged determined as functions of strain. The discrete symbols of the "averaging" set shows the obtained values from the measured values, and the continuous line represents the fitted polynomial.

### 3.2. Finding $\beta$

Figure 2a shows the plots used in estimating $\beta$, which is defined to be temperature-independent, and it is common to report only its temperature-averaged value for any given strain; however, Figure 2b shows this assumption is not valid in general, and $\beta$ can show temperature-dependence behavior. Moreover, a single value of $\beta \approx 0.13$ is obtained as the total average.

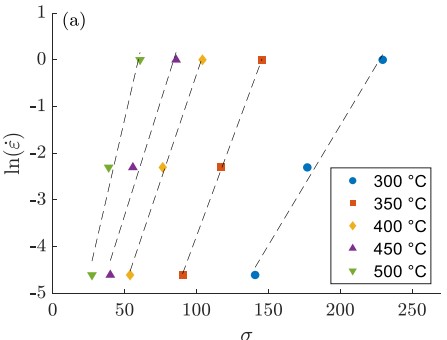
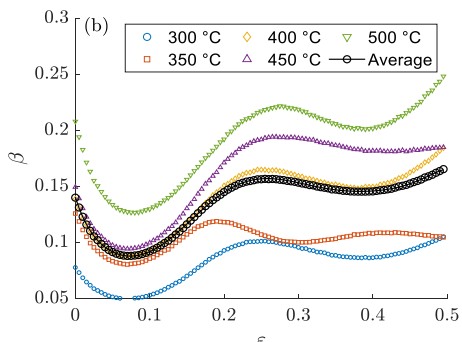

**Figure 2.** (**a**) The linear plots for determining $\beta$, and (**b**) determined values of $\beta$ and their *T*-averaged as functions of strain.

### 3.3. Calculating $\alpha$

Using the previously obtained data, one can find $\alpha = \beta/n'$. Figure 3 shows $\alpha$ as a function of strain. Alternatively, its total average is found to be $\alpha \approx 0.02$. The defined form of $\alpha$, i.e., being either a constant or a function of strain, affects the results of the following steps of the procedure.

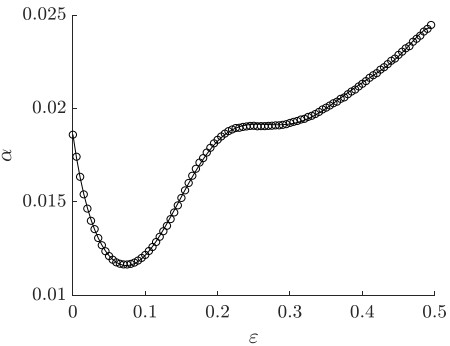

**Figure 3.** The values of $\alpha$ calculated from the *T*-averaged values of $\beta$ and $n'$ as a function of strain.

### 3.4. Finding $n$

Figure 4a shows the plots used for determining $n$, showing linear functions, yet not with similar slopes. Figure 4b shows the determined values of $n$ using two different assumed $\alpha$. The values of $n$ are functions of temperature; however, following its common definition of being temperature-independent, its average over the studied temperatures is presented here. Moreover, along the same line of the previously described constants, one can find its average over the full range of strain to be $n \approx 4.9$ for the case of $\alpha = 0.02$.

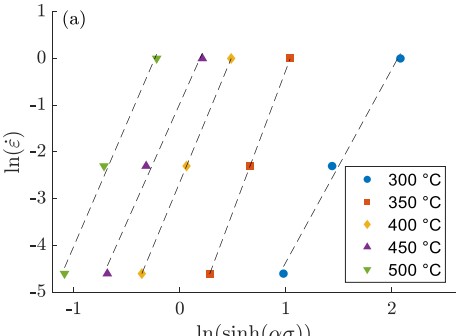
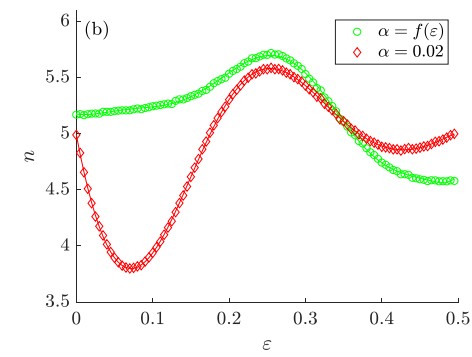

**Figure 4.** (**a**) The linear relationships for determining $n$. For this plot, $\alpha$ is defined as a function of $\varepsilon$, and (**b**) the temperature-averaged values of $n$ by imposing different values of $\alpha$.

### 3.5. Finding Q

Figure 5 shows the required plots for finding the activation energy. Even though linear regression is not the best choice here, it is common practice to approximate the dependence of this data set by linear functions. Figure 6 shows the obtained $Q$ for different strain rates as functions of strain, using both cases of $\alpha$ and $n$ being functions of strain (Figure 6a) and constants (Figure 6b). The results show their obvious different behaviors, not only for the ranges of their estimated values but also for their dependence behavior on strain: Figure 6a $Q$ shows an arbitrarily varying of the activation energy around some average value without representing the general behavior of the stress-strain curve, while in Figure 6b, $Q$ shows the general trend of the stress–strain curves, indicating a higher activation energy is required for increasing the flow stress. Assuming $\alpha$ and $n$ to be either variables or constants results in the activation energies of $\sim$192.3 kJ mol$^{-1}$ and $\sim$181.5 kJ mol$^{-1}$ for $\varepsilon = 0.5$, respectively. The latter, obtained by assuming constant $\alpha$ and $n$, is almost equal to 181 kJ mol$^{-1}$ reported by Ahamed and Senthilkumar [37].

Furthermore, Figure 6 shows that the values of $Q$ approximately have a monotonic dependence on strain rate, implying the possibility of defining $Q$ with rather a simple function of both strain and strain rate, e.g., in the form of $Q = \overline{Q}(\varepsilon) + Q(\dot{\varepsilon})$ with $\overline{Q}(\varepsilon)$ being the strain-averaged activation energies and $Q(\dot{\varepsilon})$ being an arbitrary function of $\ln(\dot{\varepsilon})$. It should be noted that the observed monotonic behavior is claimed only to be valid for the studied material, and a vast study is required to make any general statement, which is not the aim of the current work.

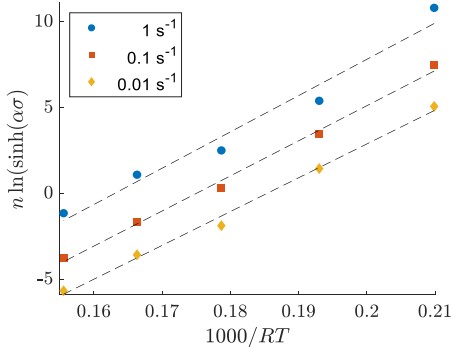

**Figure 5.** The linear plots for determining $Q$. For this plot, $\alpha$ and $n$ are considered as functions of $\varepsilon$.

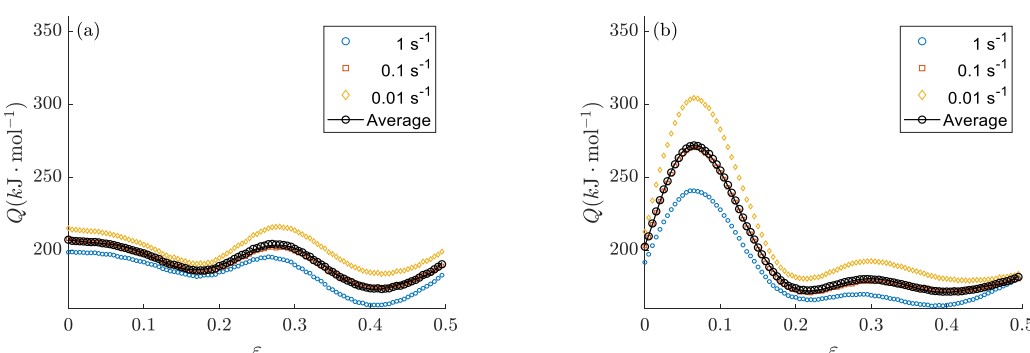

**Figure 6.** The estimated activation energies as functions of strain and strain rate, with $\alpha$ and $n$ being (**a**) functions of strain, and (**b**) constants.

### 3.6. Finding A

As described in Table 1, the last step of establishing the ST model is to find $A$ from the same plots used for determining $Q$ (Figure 5). Figure 7 shows the approximated $\ln(A)$ as functions of strain and strain rate, as well as its strain rate averaged values. Moreover, the results show that $\ln(A)$ has a monotonic dependence on $\ln(\dot{\varepsilon})$; therefore, one can define

a simple form of $\ln(A) = \ln(\overline{A}(\varepsilon)) + \ln(A(\dot{\varepsilon}))$ with its second function being an arbitrary function of $\ln(\dot{\varepsilon})$.

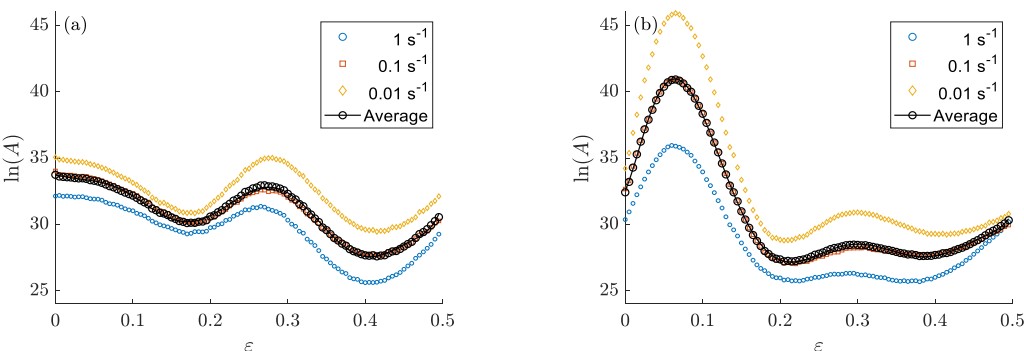

**Figure 7.** The estimated values of $\ln(A)$ as functions of strain and strain rate, for $\alpha$ and $n$ treated as (**a**) functions of strain, and (**b**) constants.

## 4. Results and Discussion

Once the parameters of Equation (2) are known, the stress corresponding to any given strain can be estimated via:

$$\sigma = \frac{1}{\alpha}\left[\sinh^{-1}\left(\frac{Z}{A}\right)\right]^{1/n}$$
$$= \frac{1}{\alpha}\ln\left[\left(\frac{Z}{A}\right)^{1/n} + \left(\left(\frac{Z}{A}\right)^{2/n} + 1\right)^{1/2}\right]. \tag{3}$$

Moreover, the performance of the model can be evaluated using different methods. In here, the mean absolute relative error (*MARE*) and the coefficient of determination ($r^2$) are used, which are defined as:

$$MARE = \frac{1}{N}\sum_i\left|1 - \frac{\sigma_{P(i)}}{\sigma_{m(i)}}\right| \text{ and} \tag{4a}$$

$$r^2 = 1 - \frac{\sum_i\left(\sigma_{m(i)} - \sigma_{P(i)}\right)^2}{\sum_i\left(\sigma_{m(i)} - \overline{\sigma}_m\right)^2} \tag{4b}$$

where $N$ is the total number of observable data, and $\overline{\sigma}_m$ is the mean value of the measured stresses, i.e., $\overline{\sigma}_m = \left(\sum_i \sigma_{m(i)}\right)/N$. The lower bound of *MARE*, that is 0, indicates a perfect match between the measured and the predicted values. A perfect correlation between the two sets of data is indicated by the upper bound of the coefficient of determination, that is $r^2 = 1$.

### 4.1. Initial Results of the ST Model

In this study, we have different sets of parameters, where $Q$ and $A$ are functions of strain in all of them. The parameters $(\alpha, n)$, however, are in four sets of $(v, v)$, $(v, c)$, $(c, v)$, and $(c, c)$, where $v$ and $c$ stand for *variable* (that is function of strain) and *constant*, respectively. Figure 8 shows the correlation between the measured and predicted stresses for the two sets of $(v, v)$ and $(c, c)$, showing some slight changes between the two models at high stress values (around the peaks). For a qualitative comparison, their performances are assessed via *MARE* and $r^2$ and summarized in Table 2. The results indicate slight improvements on reproducing the stress–strain curves by taking $\alpha$ and $n$ as functions of strain.

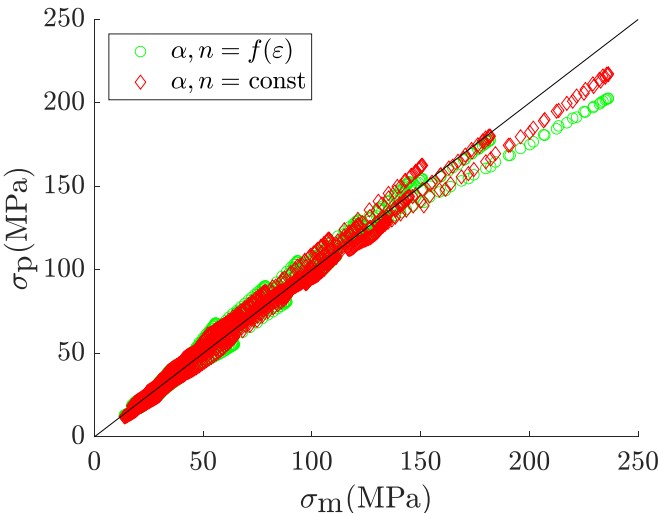

**Figure 8.** The correlation between the measured ($\sigma_m$) and predicted ($\sigma_p$) stresses using the two sets of constants, where $\alpha$, $n$ are considered both as functions of strain (green circles) and constants (red diamonds). The continuous black line represents a perfect correlation.

**Table 2.** The goodness of fit assessed by *MARE* (Equation (4a)) and $r^2$ (Equation (4b)) between $\sigma_m$ and $\sigma_p$ for the four sets of the ST model and its limiting power-law and exponential models. See Appendix A for the details of the limiting cases.

| | ST ($\alpha$,$n$) | | | | Power-Law | Exponential |
|---|---|---|---|---|---|---|
| | $(v,v)$ | $(v,c)$ | $(c,v)$ | $(c,c)$ | | |
| *MARE* | 0.066 | 0.068 | 0.068 | 0.070 | 0.059 | 0.150 |
| $r^2$ | 0.986 | 0.986 | 0.983 | 0.980 | 0.986 | 0.945 |

*4.2. Revising the ST Model*

To describe the stress values at a wide range of strains using the ST model, it is common to consider all constants of the model as functions of strain. This method seems valid at first glance, but Figure 6a shows that this method can drastically affect the estimated activation energy, both for its range and its shape as a function of strain; see [24] for a review on the topic. Most of the physical and phenomenological models (e.g., [16,20–22]) take the activation energy as their input value, which should be provided using other methods, such as the ST model; therefore, their performance depends on the accuracy of the defined activation energy, among other parameters.

Although assuming $\alpha$ and $n$ being functions of strain shows a slightly better performance of the ST model in this study (see Table 2), this assumption can result in a physically meaningless dependence of $Q$ on $\varepsilon$ in which the variations of the stress are not reflected on (see Figure 6); therefore, the values of $\alpha$ and $n$ are considered as strain-independent constants for the rest of this work, and the focus is on modifying the fitting procedure for the other two parameters, i.e., $Q$ and $A$.

4.2.1. Method 1

In this method, the first step is to define $Q(\varepsilon, \dot{\varepsilon})$. In most practices, it is assumed that the activation energy is only a function of strain, although there are some exceptions; for example, Momeni investigated the dependence of $Q$ on strain, strain rate, and temperature by separating it into thermal and mechanical parts [39]. Moreover, it is discussed that the ST model overestimates the activation energy compared to other models, e.g., Schöcks-Seeger-Wolf and Kocks model [40]. The current work is, however, limited to study the ST model.

Since the obtained activation energies show a monotonic dependence on the strain rate (Section 3.5), it can be modeled as a quadratic function of strain rate, that is $Q(\dot{\varepsilon}) = P^2(\ln(\dot{\varepsilon}))$. The results are presented in Figure 9, showing high accuracy of this estimation method.

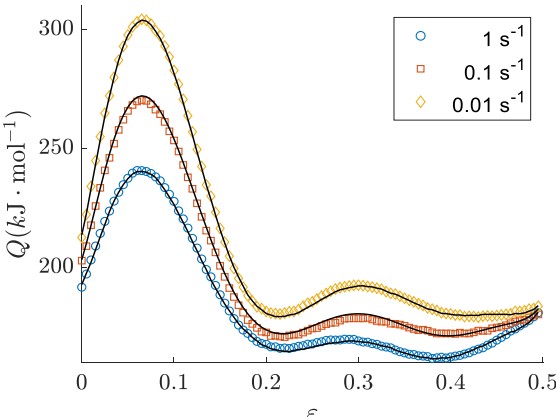

**Figure 9.** The obtained activation energies (unfilled-symbols) and their estimations (continuous lines) as functions of strain and strain rate using method 1.

The second step in this method is to define $A(\varepsilon, \dot{\varepsilon}, T)$. Once the only remaining parameter to be defined is the proportionality constant $A$, it can be obtained directly from $A = Z/(\sinh(\alpha\sigma))^n$ and expressed in terms of not only strain, but probably also strain rate and temperature. In here, a general polynomial form of $\ln(A) = P^9(\varepsilon) + P^2(\ln(\dot{\varepsilon})) + P^2(T)$ is assumed, resulting in an estimation of $A = A(\varepsilon, \dot{\varepsilon}, T)$.

In an extreme case, it is possible to assume $Q$ to be a constant instead of a function of strain, and put all the fitting weight on $A$. Regardless of the accuracy of such a fitting procedure, this approach would be unjustifiable, following its physically meaningless activation energy.

### 4.2.2. Method 2

This method is similar to method 1, but in a reverse order, i.e., $A(\varepsilon, \dot{\varepsilon})$ is defined first, and then, $Q$ is obtained from $Q = RT \ln(A(\sinh(\alpha\sigma))^n/\dot{\varepsilon})$, which can be expressed as $Q = Q(\varepsilon, \dot{\varepsilon}, T)$. Figure 10 shows some selected activation energies obtained by this method. Although the results seem promising, method 2 cannot be justified for its calculation of $Q$, because it naively converts a physically meaningful parameter to a proportionality constant.

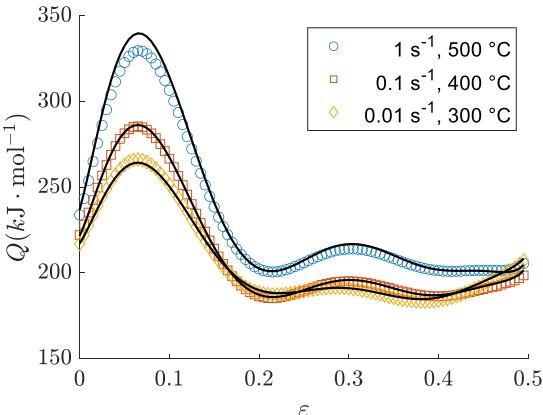

**Figure 10.** The obtained activation energies as $Q(\varepsilon, \dot{\varepsilon}, T)$ using method 2 for three randomly selected sets of strain rate and temperature.

### 4.3. The Performance of the Revised Model

The performance of the suggested methods are assessed by calculating the corresponding *MARE* and $r^2$: the summarized results in Table 3 indicate that method 1 improves the results, while method 2 decreases the accuracy of the model; see Table 2 for a comparison. It is possible to increase the accuracy of the model by allowing $\alpha$ and/or $n$ being functions of strain; however, as was discussed earlier, such assumptions result in useless activation energies.

**Table 3.** The performance of the revisited model by methods 1 and 2, assessed by *MARE* (Equation (4a)) and $r^2$ (Equation (4b)).

|  | **Method 1** | **Method 2** |
|---|---|---|
| *MARE* | 0.033 | 0.059 |
| $r^2$ | 0.995 | 0.983 |

Figure 11 shows the stress–strain curves obtained from the revised ST model (method 1) compared to the measured values. The results show the model's behavior for the whole studied range of strain, strain rate, and temperature. Figure 11c reveals some small deviations around the peak stress for $\dot{\varepsilon} = 1\mathrm{s}^{-1}$.

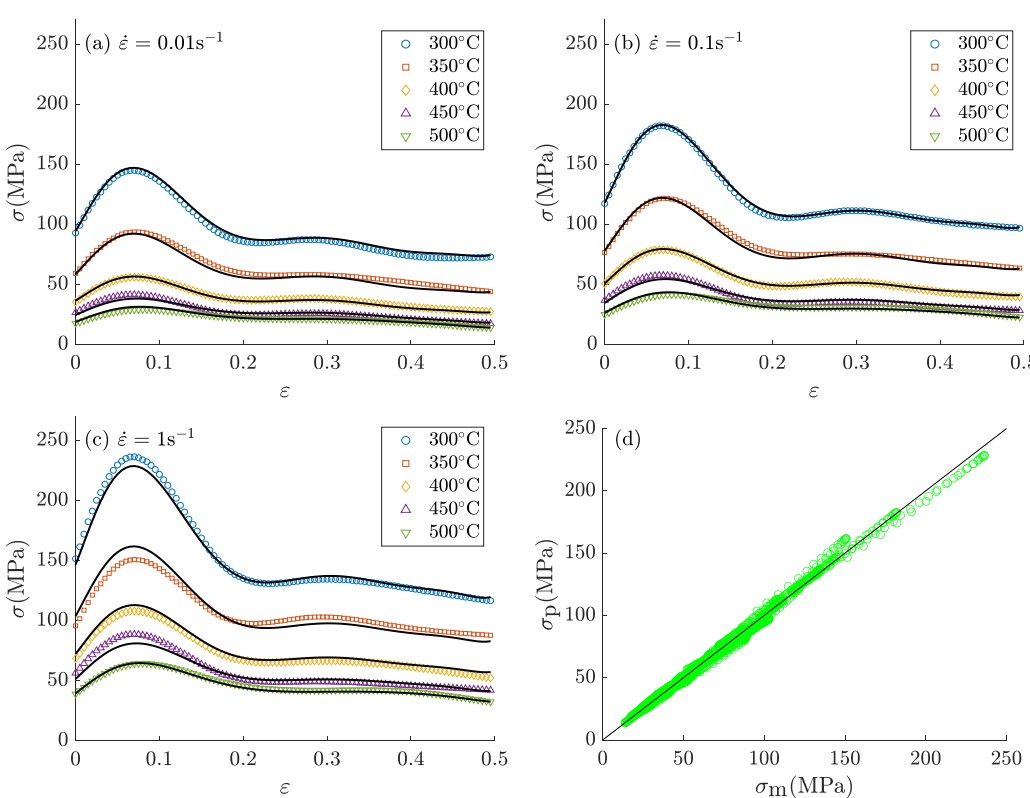

**Figure 11.** The results of the revised ST model with method 1. The stress–strain curves studied in this work at different temperatures and three different strain rates of (**a**) $0.01\mathrm{s}^{-1}$, (**b**) $0.1\mathrm{s}^{-1}$, and (**c**) $1\mathrm{s}^{-1}$. The unfilled symbols and the continuous lines represent the $\sigma_\mathrm{m}$ and $\sigma_\mathrm{p}$, respectively. (**d**) the correlation between $\sigma_\mathrm{m}$ and $\sigma_\mathrm{p}$.

## 5. Final Remarks

The current study describes the conventional method of using the Sellars and Tegart's hyperbolic sine constitutive model for modeling the stress–strain curves in hot deformations [27]. To perform the calculations of this study, a piece of code *Hot Deformation Fitting Tool* (HDFT) is written in MATLAB 9, which is published under the terms of the GNU

GPLv3 license, and can be found in https://github.com/soheilsolhjoo/HDFT (accessed on 11 November 2021) [38] .

In this work, some common practices of the ST model and their probable problematic consequences are discussed, which are summarized as follows.

1.  The first common practice is to define all constants of the model, including $n'$, $\beta$, $\alpha$, and $n$, as functions of strain. This can be achieved by taking their temperature-averaged values at any given strain. The current work reasoned that this practice can result in unjustifiable activation energies. Moreover, the results show that defining these fitting parameters as functions of strain does not effectively increase the accuracy of the model. Alternatively, it is suggested to consider these four parameters simply as constants, independent of other controlling parameters (including strain), by taking their total average values.

2.  The other common practice is to take the activation energy simply as a function of strain. Considering that the activation energy of the hyperbolic sine model is its only physically meaningful parameter, it would be reasonable to extract and express its values with more care. Aligned with this idea, a revised method for the ST model is suggested, in which the activation energy can be defined as $Q(\varepsilon, \dot{\varepsilon})$.

3.  The other assumption, made by Sellars and Tegart [27], is on considering the proportionality constant $A$ to be a temperature-independent constant; however, such assumption cannot be justified as $A$ must be free to be defined arbitrarily for its form and dependence. The current work suggests a method to express this constant of proportionality as $A(\varepsilon, \dot{\varepsilon}, T)$.

Using the suggested revised method 1 (see Section 4.2.1), in which the parameters $\alpha$ and $n$ are constants, the deformation's activation energy is $Q(\varepsilon, \dot{\varepsilon})$ and the proportionality constant is $A(\varepsilon, \dot{\varepsilon}, T)$, the stress–strain curves are approximated accurately with a mean absolute relative error of $MARE = 0.032$ and a coefficient of determination of $r^2 = 0.996$.

Considering that the data-driven models, e.g., ANNs, can describe stress–strain curves with accuracies higher than the ST and its limiting models, as is briefly discussed in Appendix B, it can be concluded that the importance of working with the ST model cannot be simply to describe a wide range of stresses anymore, but the possibility of estimating its only physically meaningful parameter, that is the activation energy of deformation. While the data-driven models are powerful tools for replicating complex nonlinear behaviors, their applications are limited to the apparent responses; therefore, relevant analytical models, such as the ST model, are required to obtain physically meaningful data, which is the activation energy of deformation in this study.

**Funding:** This research received no external funding.

**Data Availability Statement:** The data of the analyzed stress–strain curves are collected from the available literature [37]. The postprocessings are performed using HDFT [38].

**Conflicts of Interest:** The author declares no conflict of interest.

## Appendix A. The Hyperbolic Sine Model and Its Limiting Cases

*Appendix A.1. Hyperbolic Sine Model: $Z = A(\sinh(\alpha\sigma))^n$*

The hyperbolic sine function in the form of Equation (2) can be reduced to the power law and the exponential model for $\alpha\sigma \to 0^+$ and $\alpha\sigma \to +\infty$, respectively, as follows.

1.  $\alpha\sigma \to 0^+$

    This limit can be expressed by the Taylor series expansion of hyperbolic sine function:

$$\sinh(x) = \sum_{k=0}^{+\infty} \frac{x^{2k+1}}{(2k+1)!} \tag{A1}$$

$$Z \propto \left( \alpha\sigma + \frac{(\alpha\sigma)^3}{3!} + \frac{(\alpha\sigma)^5}{5!} + ... \right)^n \tag{A2}$$

$$\alpha\sigma \to 0^+ \Rightarrow \alpha\sigma \gg \frac{(\alpha\sigma)^3}{3!} + \frac{(\alpha\sigma)^5}{5!} + ... \tag{A3}$$

$$\Rightarrow Z \propto \sigma^{n'}. \tag{A4}$$

2.  $\alpha\sigma \to +\infty$

    For this case, we can use the exponential definition of hyperbolic sine function:

$$\sinh(x) = \frac{e^x - e^{-x}}{2} \tag{A5}$$

$$Z \propto \left( e^{\alpha\sigma} - e^{-\alpha\sigma} \right)^n \tag{A6}$$

$$\alpha\sigma \to +\infty \Rightarrow e^{\alpha\sigma} \gg e^{-\alpha\sigma} \tag{A7}$$

$$\Rightarrow Z \propto e^{n''\alpha\sigma}. \tag{A8}$$

The reason for changing the exponent $n$ to $n'$ and $n''$ is to compensate the truncation error, which is introduced by neglecting some terms of the models, i.e., the right-hand side of the inequities of Equations (A3) and (A7). Note that Equation (A8) can be written as $Z \propto e^{\beta\sigma}$ with $\beta = n''\alpha$. Now, if we assume the exponent of $n$ in the two limits are the same, i.e., $n'' = n'$, one can write $\beta = n'\alpha$, resulting in $\alpha = \beta/n'$. In other words, this estimation of $\alpha$ relies on combining the two limits of $\alpha\sigma$, which may seem arbitrary at the first glance; however, following this method, the initial estimation of $\alpha$ would be appropriate for the whole range of $\alpha\sigma$, and the other parameters of the model ($n$ and $A$) would compensate for this assumed estimation. Finally, it should be noted that assigning a truncation error of 10%, Equation (2) reduces to the power-law and exponential models with $\alpha\sigma < 0.8$ and $\alpha\sigma > 1.2$, respectively, as reported numerously in the literature. The suitability of the assigned error, however, has not been discussed in the literature, as far as the author knows.

In the next two subsections, both the power-law and the exponential models are discussed in detail. It should be noted that to differentiate the models used for estimating the activation energy, three different notations are introduced: $Q$ for the hyperbolic sine, $Q'$ for the power-law and $Q''$ for the exponential model.

### Appendix A.2. Power Law Model: $Z = A' \sigma^{n'}$

To use this model, its three constants $n'$, $A'$ and $Q'$ should be known. The value of $n' = 4.9$ was found in Section 3.1. Figure A1a shows the plots used for estimating $Q'$ and $A'$, and Figure A1b shows their obtained values: the activation energy is close to what was found in Figure 6a for which $\alpha$ and $n$ were treated as functions of strain. The correlation between the measured and predicted stresses are depicted in Figure A2.

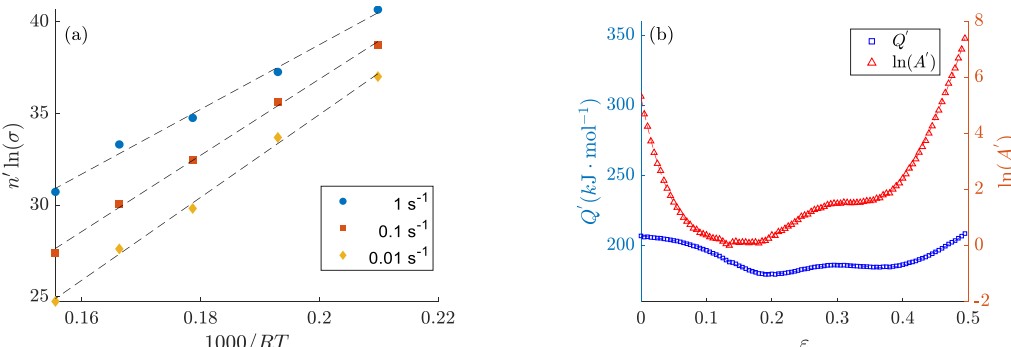

**Figure A1.** (**a**) The linear plots used for estimating the activation energy and the proportionality constant for the power law model. (**b**) The obtained $\dot{\varepsilon}$-averaged $Q'$ and $\ln(A')$.

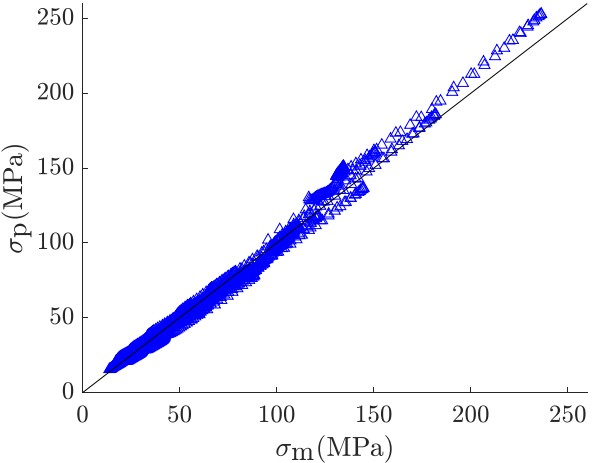

**Figure A2.** The correlation between $\sigma_{\mathrm{m}}$ and $\sigma_{\mathrm{p}}$ using the power law model.

*Appendix A.3. Exponential Model:* $Z = A'' \exp(\beta\sigma)$

This model requires the values of its three constants: $\beta$, $A''$, and $Q''$. In Section 3.2, it was found that $\beta = 0.13$. For determining the other two constants, the linear plots of $\beta\sigma$ as functions of $1/RT$ are used, as shown in Figure A3a, although the linear plots might not be the best choice here. Figure A3b shows the determined values of these two constants. The obtained values show a general behavior comparable to the results in Figure 6b, although the activation energy of the exponential model is larger than the estimations of the hyperbolic sine one. Figure A4 shows the correlation between the measured and modeled stresses.

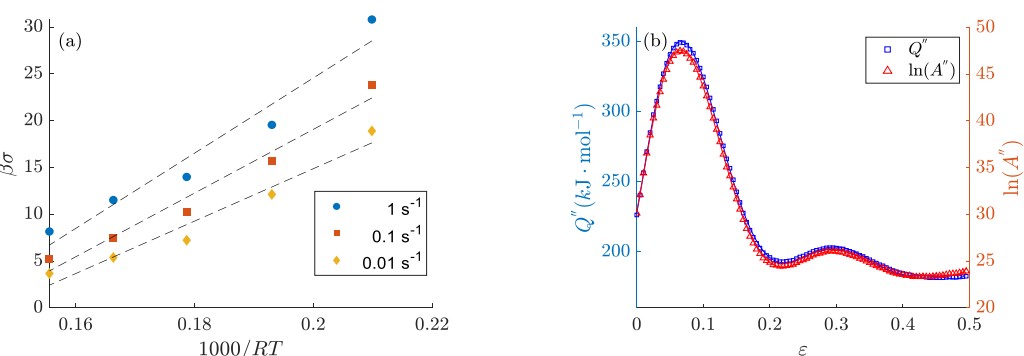

**Figure A3.** (**a**) The linear plots used for estimating the activation energy and the proportionality constant for the exponential model. (**b**) The obtained $\dot{\varepsilon}$-averaged $Q''$ and $\ln(A'')$.

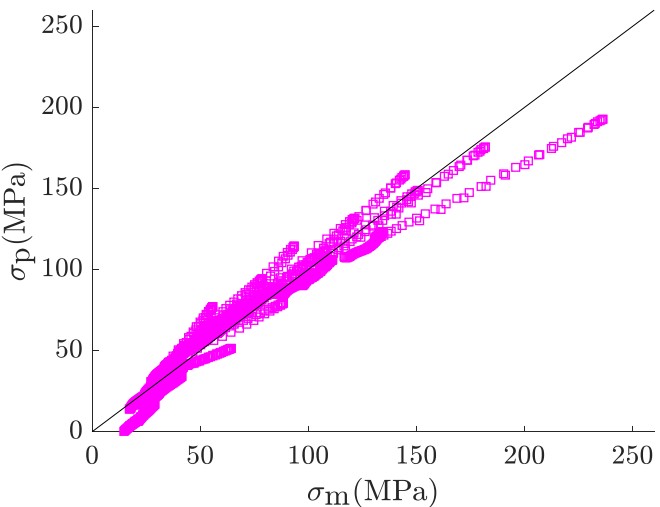

**Figure A4.** The correlation between $\sigma_m$ and $\sigma_p$ using the exponential model.

## Appendix B. Artificial Neural Networks (ANNs)

The usage of ANNs is increasing in industrial applications for establishing data-driven models. To briefly demonstrate their application in predicting stress–strain curves, a 3-5-5-1 model is used as the architecture of the neural network, that is, three neurons for the input layer (strain, strain-rate, and temperature), two hidden layers (each with five neurons), and one output (stress). The neural network is trained with Bayesian regularization backpropagation 100 times using MATLAB (trainbr) [41]. For each training process, the data points are randomly divided and distributed into training, validation, and testing sets with a ratio of 0.70, 0.15, and 0.15, respectively.

Figure A5 shows the distributions of *MARE* and $r^2$ for the trained ANNs, indicating high accuracy for the majority of the trained ANNs with their *MARE* and $r^2$ being in the ranges of $[0.01, 0.02]$ and $[0.999, 1)$, respectively. Figure A6 compares the estimated stress values for a randomly selected trained ANN and the experimental data. These results make the trained ANNs, among the studied ones in this work, to be the most accurate models for predicting the stress–strain curves. This way, the stress–strain curves are estimated with high accuracy yet without struggling to find the physically meaningful activation energy.

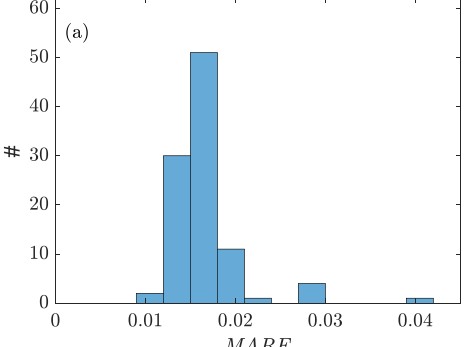
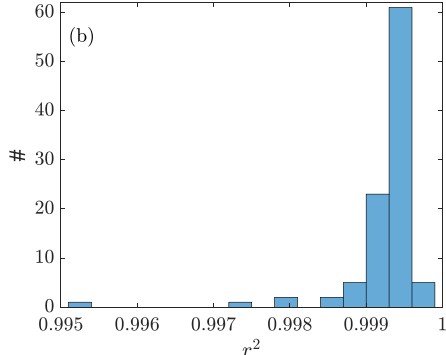

**Figure A5.** The distributions of (**a**) *MARE* and (**b**) $r^2$ for 100 trained ANNs using the Bayesian regularization backpropagation with a simple architecture: two hidden layers, each with five nodes.

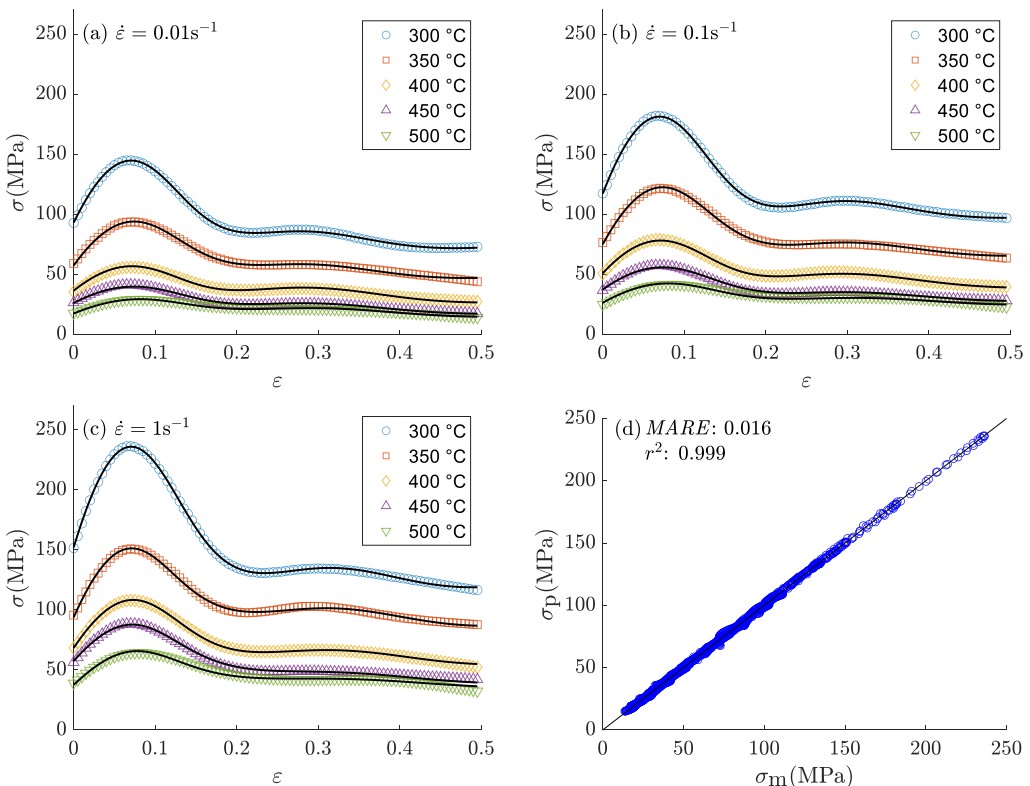

**Figure A6.** The comparison between the measured stresses and their estimations using a trained ANN. The stress–strain curves are shown for different temperatures and strain rates of (**a**) 0.01 s$^{-1}$, (**b**) 0.1 s$^{-1}$, and (**c**) 1 s$^{-1}$. (**d**) The correlation between $\sigma_m$ and $\sigma_p$.

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
