# Peer review of "Revisiting the Common Practice of Sellars and Tegart’s Hyperbolic Sine Constitutive Model"

_2673-3951, doi:10.3390/modelling3030023_

Round 1
Reviewer 1 Report
The submitted research offers various approaches to handle the well-known Garofalo’s hyperbolic-sine function with respect to a hot flow stress description. The conducted research contain some minor errors - as follows:
In the line 137, there is equation in which the letter ‘t’ in the denominator should be probably replaced by the letter ‘N’ because the ‘N’ is in the line 136 used to represent the total number of observed data. This should be checked and revised.
There is also a mistake in equation A1 in the appendix section. There is ‘2x+1’ in the exponent and also in the denominator - however, there should be 2i+1 – Please revise this. In addition, I recommend to replace the ‘i’ by e.g. ‘k’ since the ‘the term ‘2i+1’ can look like a complex number.
In the line 279, there is stated that - One prominent algorithm to train a rather simple ANN is “Bayesian regularization backpropagation”. However, the Bayesian regularization itself is only a part of the Levenberg-Marquardt (LM) algorithm. The LM algorithm is the main algorithm to get weights and biases of the ANN. The Bayesian regularization is incorporated only to prevent an overfitting issue.
Reviewer 2 Report
The author clearly described the steps that improved the application of Sellars and Tegart’s model, sharing also the MATLAB code.
I think that the article is interesting but not brings a high novelty in the literature. However, it is well-written. I have only few comments that have to be addressed before publication.
The following phrase should be modified to be clearer: “figure 6a shows that this method can drastically affect the estimated 154 activation energy as the only physically interpretable parameter of the constitutive model;”
How escluding the temperature affect the final results? see phrase: “The current work is, however, limited to study the ST model. In this section, we define the activation energy as a function of strain as well as strain rate, but not temperature.”
There are 9 self-citations (on a total of 49), I think it is a great number. A revision is necessary to maintein only the most pertinent ones.
